# A Prognostic Model for the Outcome of Nobel Biocare Dental Implants with Peri-Implant Disease after One Year

**DOI:** 10.3390/jcm8091352

**Published:** 2019-09-01

**Authors:** Miguel de Araújo Nobre, Francisco Salvado, Paulo Nogueira, Evangelista Rocha, Peter Ilg, Paulo Maló

**Affiliations:** 1University Clinic of Stomatology, Faculty of Medicine, University of Lisbon, 1649-028 Lisbon, Portugal; 2Research and Development Department, Maló Clinic, 1600-042 Lisbon, Portugal; 3Institute of Preventive Medicine, Faculty of Medicine, University of Lisbon, 1649-028 Lisbon, Portugal; 4Oromaxillofacial Surgery, University of Campinas, São Paulo 13083-970, Brazil; 5Implantology Department, Maló Clinic, 1600-042 Lisbon, Portugal

**Keywords:** dental implants, peri-implant disease, peri-implantitis, risk, epidemiology, prognosis

## Abstract

Background: This investigation, based on a 1-year retrospective cohort study, aimed to estimate and validate a prognostic model for ailing and failing implants due to peri-implant disease. Methods: A total of 240 patients (male: 97; female: 143; average age of 57.3 years) with at least one ailing or failing implant were included: 120 patients for model derivation and 120 patients for model validation. The primary outcome measure was the implant status: success, defined as the arrest of the disease, or failure defined as implant extraction, prevalence or re-incidence of peri-implant disease). Potential prognostic risk indicators were collected at the baseline evaluation. The relative risk (RR) was estimated for the predictors through logistic regression and the c-statistic (95% confidence interval) was calculated for both derivation and validation sets. The significance level was set at 5%. Results: The risk model retrieved the prognostic factors age (RR = 1.04), history of Periodontitis (RR = 3.13), severe peri-implant disease status (RR = 3.26), implant length (RR = 3.52), early disease development (RR = 3.99), with good discrimination in both the derivation set (0.763 [0.679; 0.847]) and validation set (0.709 [0.616; 0.803]). Conclusions: A prognostic risk model for estimating the outcome of implants with peri-implant disease is available, with a good performance considering the c-statistic evaluation.

## 1. Introduction

Dental implant treatment has evolved over the years to a predictable outcome in the long term, with high success rates reported irrespective of the type of restoration (single teeth, fixed partial prosthesis or full-arch) [1,2,3]. Nevertheless, mechanical and biological complications can occur, threatening long term implant success [1]. One of such complications is peri-implant disease.

Defined as the term for inflammatory reactions with loss of supporting bone tissue surrounding an implant in function [4], influenced by biological and biomechanical factors that negatively affect the outcome of implant supported restorations with presence of inflammatory signs and loss of supporting bone around an implant in function [5,6], the prevalence of peri-implant disease is reported in the literature with a wide range, between 4.7% to 43% at the implant level, and from 1% to >56% at the patient level [7,8,9,10,11]. This observed variability in the prevalence of the condition between different studies is at least partly explained by methodological issues related to heterogeneous case definitions and diagnostic criteria, making it challenging to compare across studies. Nevertheless, recent systematic reviews and meta-analysis investigating the frequency of peri-implant diseases reported a frequency/prevalence of 18.8%–22% for peri-implant disease at the patient level [8,10,12].

From a causal component mechanism standpoint, as a chronical condition, peri-implant disease is potentially influenced by multiple risk indicators including: history of Periodontitis, biological factors such as biofilm, bleeding and bone level, biomechanical factors such as occlusal overloading, prosthesis lack of passive fit, prosthetic component loosening, type of material used in the restoration or the proximity of other implants and/or teeth, and background factors related to poorly constructed implants placed and handled by untrained clinicians, together with patient characteristics and function time [6,10,13,14,15,16,17,18]. Considering peri-implant disease as the major risk for late implant failure, and given that the outcome of therapy of implant-supported rehabilitations affected with peri-implant disease is considered to be unpredictable [5], it is important to develop strategies that can help avoiding the incidence of the disease: In this sense, primary prevention is paramount [19], with patient self-care and clinical maintenance assuming an 0 important role as effective preventive measures to manage peri-implant diseases [8,20]. Moreover, the use of risk scores that help identifying patients at a greater risk for the incidence of disease represent an important tool, simplifying the assessment of the multi-factorial nature of peri-implant disease for the clinician, while motivating patients to change risk factors in order to reduce their own risk [6].

Nevertheless, when peri-implant disease is diagnosed, the outcome is not predictable and in the majority of situations it is not possible to arrest progression, with more than 50% of disease re-incidence or implant failure [20]. The study of prognostic factors in patients already affected by the disease, especially in multivariable analysis (prognostic models) represent an important tool used for some time in other specialties of Medicine to increase the predictability of the outcome or to properly select patients for interventions [21,22]. However, models for patients with peri-implant disease are absent. The aim of this study was to report on the derivation and validation of a prognostic model for implants with peri-implant disease.

## 2. Materials and Methods

This article was written following the Strobe (Strengthening the Reporting of Observational Studies in Epidemiology) guidelines [23]. This retrospective cohort study was approved by the Ethical Committee for Health (authorization 010/2013), the Faculty of Medicine-University of Lisbon Ethical Board (Process 270/2015) and the Faculty of Medicine-University of Lisbon Scientific Board (Process CC-120), following the Helsinki Declaration of 1975, as revised in 2000. The patients provided an informed consent in order to participate in the study. 

The study was conducted at a private rehabilitation center (Lisbon, Portugal) for the clinical evaluations and at the Faculty of Medicine, University of Lisbon (University Clinic of Stomatology) for the data analysis and manuscript preparation; between March 2015 and August 2017. The patients were rehabilitated between January 2012 and December 2015 and followed until December 2016; Data collection occurred between January 2016 and November 2016; Data analysis occurred between November 2016 and August 2017. 

The population included patients over 18 years, of both genders, rehabilitated with the same type of implants at a center for Implantology and Fixed Oral Rehabilitation (Lisbon, Portugal). The patients were randomly selected from the archives of an epidemiological surveillance study [11] at the private rehabilitation center. Inclusion criteria were patients rehabilitated with implant supported fixed prostheses (all implants placed using a non-submerged surgical protocol) with a diagnostic of peri-implant disease, with a minimum of 1 year post-loading (cut off value to prevent overlapping between early implant failure and peri-implant disease); followed at the clinical center and who agreed to participate through written informed consent. The exclusion criteria were patients who refused or were unable to give informed consent; whose medical records were incomplete/missing; patients not followed at the private rehabilitation center; patients presenting with peri-implant pockets >4 mm in the first year of implant function (in order to comply with the healing adaptive principle [24] and not to overlap between early failure and peri-implant disease), and patients with bone level located on the implants’ apical third.

Peri-implant disease was defined as the presence of peri-implant pockets ≥5 mm, bleeding on probing and/or suppuration, with concurrent presence of marginal bone loss or clinical attachment loss ≥2 mm comparatively to the previous examination) [25]. The disease severity was defined using a threshold for marginal bone loss at the time of diagnosis: a radiographic image exhibiting a marginal bone loss of more than 1/3 of the implants’ length was defined as severe peri-implant disease, while a marginal bone loss of less than 1/3 of the implants’ length was considered as moderate peri-implant disease. 

Baseline recordings were set at the diagnosis of peri-implant disease. The patients’ records were reviewed: record sheets, radiographs, medical history and data collection, filling in the data on a digital form. Examinations were performed by a team of calibrated clinicians with estimation of inter-examiner reliability [11]. The dental implants inserted in this study were from the Nobel Biocare system (Brånemark system, NobelSpeedy, Nobel Biocare AB, Gothenburg, Sweden) and the same team performed implant insertion and surgical treatment on the implants. 

### 2.1. Follow-Up and Interventions

The patients were followed from baseline (time of diagnosis of the disease) with clinical observations at the time of disease recurrence or at 2 months, 6 months and one year of post-diagnosis. In these clinical appointments, beside the diagnostic measures the patients received prophylactic maintenance and self-care indications. The prostheses were removed at each clinical appointment.

### 2.2. Predictors and Model Development

The dependent variable was the implant prognostic (remission or failure): The state of remission was defined as the maintenance of a probing pocket depth ≤4 mm, and absence of suppuration, bleeding on probing, mobility (evaluated manually) or marginal bone loss (evaluated through standardized radiographs) during the period of one year post-diagnosis; the state of failure was defined as the removal of the implant; the maintenance, progression or re-incidence of the disease within one year of the post-diagnostic follow-up, characterized by implant mobility, or the presence of peri-implant pockets >4 mm and bleeding on probing/suppuration, or additional marginal bone loss (evaluated through standardized radiographs) [26,27]. 

Patient and restoration related characteristics that could be determined easily and reliably during follow-up appointments were considered for inclusion in the prognostic model. If a patient registered two or more implants with peri-implant disease, one implant was randomly selected. Initially, a set of 20 potential predictor variables associated with good or bad prognosis of peri-implant disease were examined (all collected at the point of diagnosis) [28,29,30,31,32,33]. The selection of 20 variables a priori for the univariable analysis was performed considering the fulfilment of a number of criteria for causal inference [34]: temporality, (the exposure preceded the effect), biological plausibility (the cause and effect story made biological sense), coherence (the cause and effect story made sense considering all knowledge available). These predictors included demographics (age in life years and gender); history of periodontitis (presence; absence), comorbidities (presence; absence), smoking habits (non-smoker; current smoker), implant position per arch (maxilla; mandible); implant position within the arch (anterior; posterior); proximity of the implants to other teeth or implants (absence of an implant or natural tooth immediately next to the implant; implant localized next to natural teeth or other implants); early disease development (time of follow-up in months at the point of diagnostic recoded to: <4 years; ≥4 years); implant length (measured in millimeters and recoded to: ≤13 mm; >13 mm); implant surface (machined, anodically oxidized); abutment height (1 mm, 2 mm, 3 mm, 4 mm, 5 mm, and 6 mm); type of opposing dentition (removable prosthesis; natural teeth; fixed prosthesis over natural teeth; implant-supported prosthesis); type of material used in the restoration (full ceramic, metal-ceramic, metal-acrylic, full acrylic resin); incidence of mechanical complications (presence or absence of fractures or prosthetic component loosening); bioflim based on the modified plaque index (mPLI) [35] (absence or presence); bleeding based on the modified bleeding index (mBI) [35] (absence or presence); peri-implant pocket depth measured in millimeters using a plastic periodontal probe (Click-Probe, Kavo, Bioggio, Switzerland) calibrated to 0.25 N (≤6 mm; >6 mm); disease severity evaluated through marginal bone loss (located at the implants’ coronal third or middle third); therapeutic approach for the treatment of peri-implant disease (non-surgical treatment alone; combination of non-surgical and surgical treatments). For the present analysis the predictors that were related to the outcome were selected based on multivariable analysis considering the remaining criteria for causal inference: strength of association, consistency, specificity, biological gradient, experiment and analogy [34]. The prognostic model was derived from the first set of patients that were randomly selected (derivation set) and was again tested in a second set of patients that were randomly selected (validation set). 

### 2.3. Bias

Strategies to prevent or limit bias included the diagnosis through several clinical and radiographic criteria and including only implant restorations with at least 1 year of follow-up to prevent diagnostic error. Primary care patients from the same population were selected (no referral patients were included), reducing the probability of population bias. 

Potential confounding of biofilm and smoking was tested statistically evaluating the association as a prognostic factor and the association with other potential prognostic factors. 

### 2.4. Study Size

The sample size was calculated using software (Power and Sample size calculation, v.2.1.30, February 2003) [36], considering significance level of 5%; statistical power of 80%; a probability of failure of 50% [20] and a relative risk of failure of 1.54. Considering these calculations, 100 patients should be included. Considering a potential patient lost to follow-up rate of 20%, 120 patients were included. Two sets of patients were selected, a first set of 120 patients to derive the potential prognostic model and a second set of 120 patients to test the predictive capacity of the prognostic model. 

### 2.5. Statistical Methods

Descriptive statistics and multivariable inferential statistics were performed using the patient as unit of analysis. The characterization of the sample was performed through frequencies and descriptive statistics to the demographic variables (age, gender). Univariate logistic regression analysis was conducted to test the association of potential predictors and failure. The variables individually associated with the outcome of failure (*p* < 0.05) were tested in a multivariable logistic regression model. The statistical performance of the final model was evaluated according to the variables’ statistical significance (*p* < 0.05) and the Omnibus tests of Model Coefficients (chi-square tests to test the improvement of the model with the explanatory variables compared to the baseline model). The inclusion of risk indicators in the model was performed considering the potential statistical limitations of inserting too many variables, yielding a statistical bias (10 events per variable) [37]. 

With this method it became possible to identify the variables that maintained their weight in association with the prognosis of peri-implant disease when controlled for other factors. For each variable identified as independent factor, a coefficient β, *p*-value of the test, and relative risk (RR) estimate with 95% confidence interval (CI) were given. The agreement or difference between predicted and observed event (calibration) was assessed using the Hosmer-Lemeshow chi-square estimates. The level of significance chosen was 5%. Statistical analysis was performed using the software IBM SPSS version 17 (IBM, Chicago, IL, USA).

### 2.6. Internal and External Validation

The discriminatory power of the model in the derivation set (internal validation) was evaluated through the area under the curve statistic (AUC) using receiver operating characteristic (ROC) curve. The ROC curves were estimated indicating the AUC and 95% confidence intervals (95% CI). The final model was tested in the validation set (external validation), a new random sample of 120 patients with peri-implant disease. Performance criteria comprised the discriminatory power measured using the AUC. 

### 2.7. Model Presentation

The final model was presented in a score chart, using the scores based on the regression coefficients (beta values) and the transformation of the logit to probability using a logit transformation table (*p* = 1/[1 + e^−logit(*p*)^]). The relation between risk indicators and outcome can be calculated from an Excel spreadsheet.

## 3. Results

### 3.1. Participants 

There were initially 1711 patients diagnosed with peri-implant disease. Of these, 428 patients were excluded due to representing long term prevalent cases and 5 patients with implants inserted in another center. From the remaining 1278 patients, 49 had the clinical and radiographical manifestations of the disease during the implants’ first year of follow-up and were excluded, rendering a final 1229 patients potentially eligible for inclusion. During the random sampling process, 27 patients registered bone level on the implants’ apical third and were considered as failures not included in the final 240 patients selected for inclusion in the present study. Therefore, the current analysis was based on data from 240 patients with 1590 implants (with one implant per patient randomly selected). 

Patient characteristics are depicted in Table 1. A total of 73 female and 47 male patients composed the derivation set, while 69 women and 51 male patients composed the validation set. The average age of the patients in both the derivation and validation sets was 58.6 years. One patient from the validation set (0.8%) was lost in the follow-up.

### 3.2. Results Reported According to Disease Severity

Five patients had five implants removed during follow-up because of persisting peri-implant disease and 126 additional patients had their implants classified as failures based on the previously stated criteria. In total, 131 patients (54.6%) showed failure in remission of the disease. The outcome of peri-implant disease after 1 year of follow-up was characterized by a failure percentage of 54.2% (*n* = 65 patients) and 55% (*n* = 66 patients) in the derivation and validation sets, respectively. Disease severity significantly influenced the outcome of peri-implant diseased implants after 1 year of follow-up: Severe disease led to poor prognosis in both the derivation and validation sets, with 63.6% and 62.1% prevalence of disease, respectively, compared to 42.6% and 49.2% with moderate disease status in derivation and validation sets.

### 3.3. Treatment Protocols

The treatment protocol included non-surgical treatment, alone or in combination with surgical treatment: Non-surgical treatment alone was performed (*n* = 197 patients; derivation set: *n* = 100; validation set = 97 patients) by scaling with an ultrasonic scaler device, irrigating with 0.2% chlorhexidine gel and prescribing the same solution for the patient to administrate at home in daily care, followed by a re-evaluation after two months. The surgical treatment was performed (*n* = 43 patients; derivation set: *n* = 20 patients; validation set = 23 patients) through open flap, removal of granulation tissue, decontamination of the implant surface both mechanical using an ultrasonic scaler and chemically through a polish using a brush attached to the contrangle and 0.2% chlorhexidine and saline irrigation. Subsequently, the mucosal flap was apically positioned, sutured with 4-0 non-resorbable sutures (Braun Silkam non-absorbable 4-0, Aesculap, Tuttlinged, Germany), and systemic antibiotics were prescribed to the patient (amoxiciline 875 mg + clavulanic acid 125 mg, Labesfal, Campo de Besteiros, Portugal) starting three days pre-surgery and extending to five days post-surgery. Suture removal occurred after 10 days and a re-evaluation after two months.

### 3.4. Prognostic Factors

The results of the univariable and multivariable logistic regression analysis are presented in Table 2. The results of the univariable analysis showed a significant association (*p* < 0.05) between the outcome (implant failure or disease progression after 1-year of follow-up) and five prognostic factors. The RR of the five prognostic factors included in the multivariable prognostic model were: age (RR = 1.04, 95% CI [0.99–1.08], *p* = 0.105), history of periodontitis (RR = 3.13, 95% CI [1.15–8.55], *p* = 0.026), severe peri-implant disease status (RR = 3.26, 95% CI [1.37–7.81], *p* = 0.008), implant length > 13 mm (RR = 3.52, 95% CI [1.48–8.37], *p* = 0.004) and early disease development (RR = 3.99, 95% CI [1.62–9.82], *p* = 0.003). 

The AUC of the model was 0.763 (95% CI [0.679; 0.847], *p* < 0.001) and is illustrated in Figure 1.

### 3.5. Model Presentation

A simple score chart for the application of the prognostic model is presented in Figure 2, Figure 3 and Figure 4, which can be used to obtain approximations of the relation between risk indicators and outcome for individual patients. For example, a 65 years old patient with history of periodontitis, registering peri-implant disease in an implant of 11.5 mm of length with less than 4 years of follow-up at diagnostic and bone level located on the implants’ coronal third, has a model score of 1.299 (log odds). According to Figure 3, this score corresponds to poor prognosis with the risk of unfavorable outcome of approximately 80% at one-year post-diagnosis.

### 3.6. Prognostic Model Validation

The prognostic model maintained the significance (*p* = 0.007; Omnibus test of model coefficients) and calibration (*p* = 0.211; Hosmer and Lemeshow test) in the validation set. The discriminatory ability of the model slightly decreased in the validation set, with an AUC of 0.709 (95% CI [0.616; 0.803], *p* < 0.001) as illustrated in Figure 5. 

## 4. Discussion

This study investigated potential prognostic factors for peri-implant disease. The multivariable prognostic model included the variables: age, history of periodontitis, severe peri-implant disease status, implant length >13 mm, and early disease development. The prognostic model registered a good discrimination ability to correctly predict the one-year post-diagnostic outcome of implants with peri-implant disease. 

In the present study, the term “peri-implant disease” is used to classify the chronical pathology affecting dental implants in replacement of the term “peri-implantitis”. The currently used terminology “peri-implantitis” has been challenged previously as it represents a term used to imply “a disease process similar to periodontitis, filling the vacuum created by our incomplete understanding of the implant-host interface with our reasonably coherent understanding of the tooth–host interface/periodontium” [38]. The use of this definition is supported by recent genetic investigations that have revealed differences in gene expression between patients with peri-implant disease and those with periodontitis [39,40]. The etiology, which is considered to be a pathology largely produced by a bacterial infection of plaque accumulation [41], was previously refuted [6,15,42,43,44] with several factors aside from biofilm emerging as potential risk factors in a multivariable analysis [43].

The present study has strengths and limitations. The main strength of the present study is related to the scarce existence of studies constructing short-term prognostic models for implants with peri-implant disease [44] and inexistence of models validated in a different sample than its derivation. The innovative results can therefore be used to help in the decision-making process in the attempt to discriminate between good or poor prognosis of implants with peri-implant disease. Furthermore, the use of a cohort study design; a multivariable approach to design and analysis (as usually individual risk assessment is poor when based on just one risk indicator); the well-defined population; reporting the results according to the disease severity and including disease severity in the prognostic model controlling for the potential bias; the uniformity with cases from the same center using standardized protocols (for diagnostic, interventions and follow-up periods) represent further strengths [45,46,47]. Nevertheless, despite the standardized protocols, the attribution of a treatment modality (namely surgical intervention) did not depend only on local factors present at the implant site but also on the patient’s acceptance, leading to variations in the use of the surgical intervention and representing a limitation. The formal randomization process during sampling aimed to address this issue, and despite the non-significant association between the variable surgical intervention and the prognosis of the implant this result should be interpreted with cation considering the limitation described. A second limitation relies on the evaluation of clinical mobility that was performed manually, preventing its use at a quantitative level that could further enhance the outcome measurement if resonance frequency analysis systems were used. Other variables such as the presence/absence of attached gingiva around the implant or cemented vs. screwed restorations were not investigated considering the premise that these factors needed to be addressed on the treatment phase (and therefore with potentially limited prognostic value) and on the basis that it would not be possible to study all variables, which represents a limitation. Another limitation relies on the fact that the current study was performed in patients with dental implants from the Nobel Biocare system, which advises caution in the generalization of the results to other dental implant systems. A further limitation was the fact that the present study was conducted in a rehabilitation center with several units but at which the majority of patients were Portuguese. Although validated, prospective studies at multiple centers and with heterogeneous populations are needed in the future to further refine the model.

Prognosis consists in the estimation of the risk of a future outcome in individuals based on their clinical and non-clinical characteristics [46]. Despite the inclusion of prognosis as a principal concept of medicine by Hippocrates, prognostic research receives limited attention when compared with therapeutic and etiological research [46], as it is virtually inexistent in Implant Dentistry. Prognostic factors consist in clinical, molecular or pathologic characteristics of a patient or disease (a starting point) that are associated with a clinical outcome (an end point) [45,48]. Hence, prognostic factors discriminate groups of patients with a different average prognosis and thus inform and enhance the basic prognosis [45], being used to identify the likely progression of a patient’s disease. 

This is the first study to derive and validate a model to quantitatively estimate the risk of unfavorable outcome in patients with peri-implant disease through a prognostic model. The largest amount of prognostic information was contained in the core of the five predictors included in the model: age, history of periodontitis, disease severity, implant length >13 mm and early disease development. The predictors in our model are consistent with previously reported predictive markers associated with treatment outcome of implants with peri-implant disease namely: early disease development (disease diagnosed before four years of follow-up) [26] and bone loss [27,49], both significantly associated with failure of peri-implant disease treatment. Charalampakis et al. [26] in a follow-up study of 245 patients after treatment of peri-implant disease, registered a follow-up under 4 years with failure of peri-implant disease treatment, being the only variable that remained significantly associated with the outcome in multivariable analysis. Considering the quantitative analysis of the present study, early disease development was the most significant prognostic factor based on the relative risk. Lagervall and Jansson [49], in a retrospective study to investigate the treatment outcome of peri-implantitis and to identify factors influencing the treatment success rate in 150 patients, registered implants with bone loss exceeding 1/3 of the implant length to be significantly associated with failure of treatment. De Waal et al. [27] in a study investigating prognostic indicators for surgical treatment of peri-implant disease after 12 months in 74 patients, associated an average bone loss of 4 mm at baseline with a poor outcome. Disease severity was expected to significantly influence the prognostic of implants with peri-implant disease, as lower bone levels correspond not only to less implant capacity to respond to treatment but also for sustaining occlusal load, decreasing the probability of a good outcome. Early diagnosis of peri-implant disease is therefore essential to increase the probability of a good prognosis.

History of Periodontitis was identified as a prognostic factor in the present study and was previously reported to be a risk indicator for peri-implant disease [41,43]. Patients with history of periodontitis that received dental implants have previously demonstrated a tissue-destructive inflammatory response in the periodontium, with an exacerbated inflammatory response that may be activated in the development of peri-implant disease [50]. This mechanism may explain the fact that one in every four healthy individuals developed peri-implant disease, compared to 1.7 in each individual with history of periodontitis [51], representing a potential predisposing factor [43]. This inflammatory burden may also be the mechanism influencing a poor prognosis of implants with peri-implant disease.

Implant length (implants >13 mm) were registered as a prognostic factor for poor outcome. This result should be interpreted within the context of the implants’ macro and micro designs. Bone loss was measured in a standardized approach using the implant’s coronal, middle or apical thirds, but this approach might give different absolute values of bone loss between implants of 7 mm of length (where the implant’s coronal third would comprise up to 2.3 mm) and implants of 18 mm of length (where the implant’s coronal third would comprise up to 6 mm). Dental implants with peri-implant disease imply the presence of bone loss and attachment loss, exposing the implant surface to the oral environment through soft tissue recession or inducing the formation of peri-implant pockets, both potentially favoring the accumulation of biofilm. In the presence of these conditions, the effective removal of biofilm might be more difficult. Considering the micro design, the majority of implants in this sample was of anodically oxidized surface which according to previous studies on prognostic markers for the treatment of peri-implant disease implies a potential favorable environment for biofilm accumulation in the presence of poor oral hygiene habits [27], with bacteria expressing a more virulent and resistant profile in a mechanism of acute infection at early stage whose remission to healthy levels may be difficult, considering the current treatment alternatives [26]. 

In the present study, post-hoc analysis revealed that the distribution of biofilm according to the implant length was significantly different between implants of up to 13 mm of length and implants of more than 13 mm of length: patients with longer implants exhibited more plaque accumulation, a condition that might negatively influence the prognosis and should be further investigated in new studies.

Age was significantly associated with a poor prognosis after one year of follow-up on the univariable analysis but not in the multivariable analysis when controlled for other variables of interest. The effect of age as a risk indicator for peri-implant disease was previously reported in clinical studies [11,52,53,54]. With increased age, a higher incidence of chronical systemic diseases is common with a potential effect of compromising directly or indirectly peri-implant health. This result may be associated with lower immunity and compromised healing capacity, contributing to the development of peri-implant disease when associated with history of periodontitis and cardiovascular disease and increasing the probability of a poor outcome [55]. Nevertheless, considering the prognosis of peri-implant disease, age might have secondary prognostic significance when the remaining factors are controlled, judging by the results of studies evaluating prognostic markers in the treatment of peri-implant disease (age with a non-significant association with failure) [26,27], and confirmed by the results of the present study.

There are strengths and limitations in the use of these prognostic factors on the clinical setting: The strength of these particular prognostic factors relies on their determination being performed easily and reliably during follow-up appointments. The potential limitations may rely on the prognostic factor history of periodontitis that in complete edentulous patients is dependent on the patients’ memory and the short follow-up time providing an insight of the prognosis.

Several previously reported risk factors or potential confounders for peri-implant disease, including smoking and biofilm, were also considered in our study, but were consequently not included in the final model. While biofilm is still considered as an unproven cause of peri-implant disease [56,57], previous literature suggested that biofilm and smoking could potentially be a confounder for the incidence of peri-implant disease as they related to periodontitis [14,43]. Nevertheless, statistical analysis was performed in the present study to investigate the hypothesis of a confounding effect, with the results failing to demonstrate a significant association of biofilm or smoking with the outcome or any of the predictors used in the prognostic model. Furthermore in the current setting, smoking could be considered to be a potential predicate factor of treatment response rather than a prognostic factor. Despite the facts it seems advisable to recommend patients for smoking cessation and improvement of oral hygiene habits as part of the strategy to resolve peri-implant disease [26].

The prognostic model was validated in a second sample of 120 patients (validation sample). The results obtained in the prognostic model indicate a high degree of significance (*p* < 0.001 for the goodness of fit), together with a good discrimination ability for unfavorable outcome given by the AUC statistic above 70% in both the derivation set (0.7) and validation set (0.709). This qualitative interpretation is performed based on the limits of the AUC that range between 0.5 (no discrimination) and 1.0 (perfect discrimination) of patients with remission of the disease or patients with progressive disease. A value of the AUC between 0.7 and 0.8 (as registered in the present study) implies considering the model to have good discrimination [58,59,60], given that values rarely exceed 0.8 for risk estimation [52]. This validation is important for reliable application of models outside the respective development settings. As this represents the first prognostic model for implants with peri-implant disease, the model should be tested in other samples and populations in order to increase its external validity or may serve as a reference for the comparison of new model evaluating the prognosis of implants with peri-implant disease.

Given that peri-implant disease is a chronic disease, re-treatment may be necessary, and therefore larger well-designed RCTs with follow-ups longer than 1 year are needed [61]. The authors propose the current prognostic model to aid in the selection of patients for more efficient design of clinical trials in the investigation of treatment outcomes in patients with peri-implant disease, for example by excluding both patients with very poor or very good prognosis while performing covariate adjustment of a treatment effect. 

The present study represents the first step of a long term evaluation of the outcome of peri-implant disease that is planned to be investigated at 5 years post-diagnosis, with the current model expected to be re-evaluated at 3-years and 5 years post-diagnosis. 

## 5. Conclusions

Within the limitations of this study, a prognostic model that provided good discrimination between patients with good and poor outcome of peri-implant disease after 1-year of diagnosis was computed. This model may be useful for providing representative information to patients with peri-implant disease on outcome expectations, for stratification and covariate adjustment in clinical trials, or as a topic for evaluating the quality of care.

## Figures and Tables

**Figure 1 jcm-08-01352-f001:**
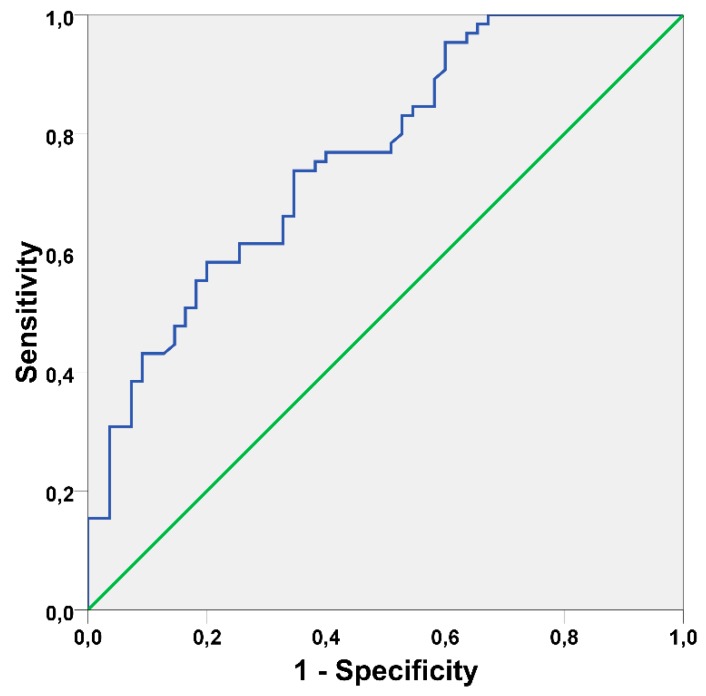
Receiver operating characteristic curve illustrating the performance of the prognostic model on the derivation set (Area under the curve = 0.763 95% CI [0.679; 0.847], standard error = 0.043, *p* < 0.001).

**Figure 2 jcm-08-01352-f002:**
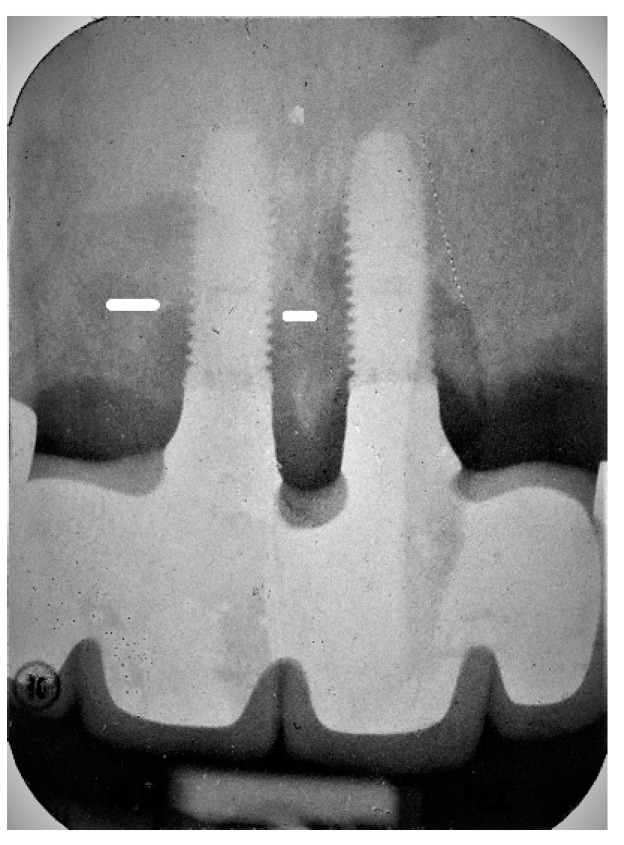
Baseline periapical radiograph at disease diagnosis of implant #11 with bone level located on the implant’s 5th thread.

**Figure 3 jcm-08-01352-f003:**
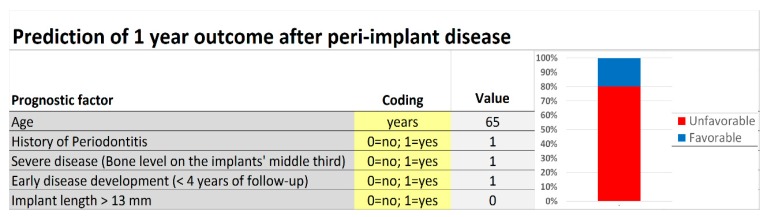
Screenshot illustrating the calculation of probability for the association between the risk indicators and the outcome. The risk assessment was calculated for a 65 years old patient with history of Periodontitis, severe disease state with bone level located on the implants’ middle third, early disease development with diagnosis before the 4 years of follow-up, and the implant with 11.5 mm of length. The prognostic model yielded an unfavorable prognostic with 80% probability.

**Figure 4 jcm-08-01352-f004:**
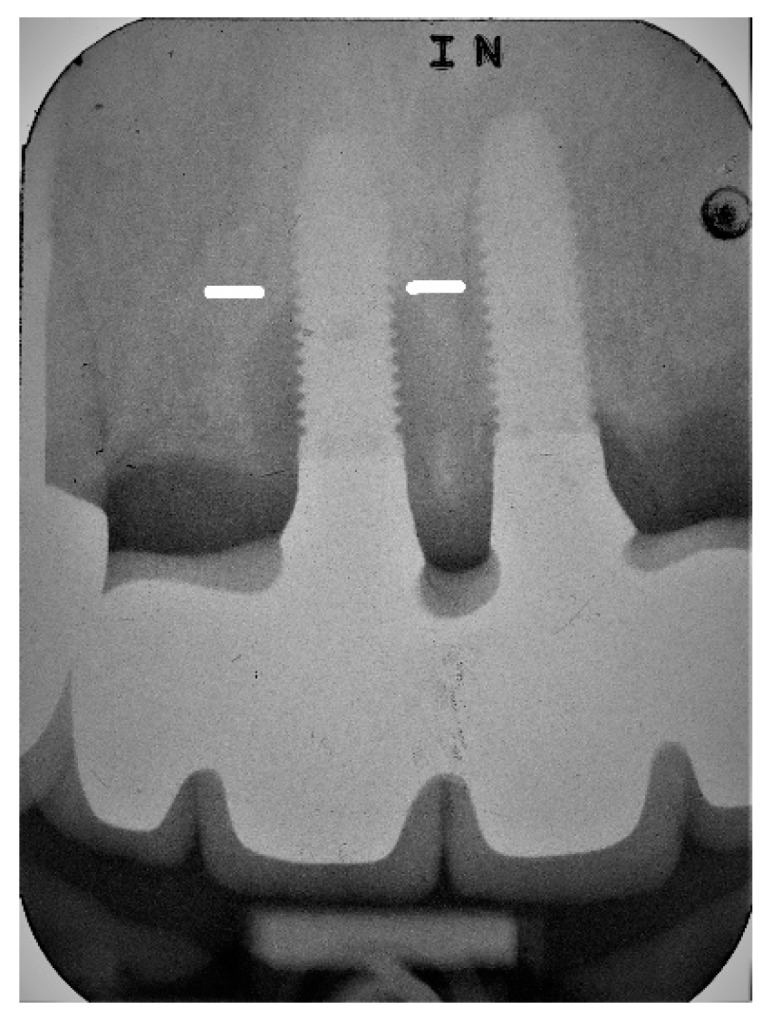
Evaluation periapical radiograph of implant #11 exhibiting further progression of the disease with the bone level located on the implant’s 9th thread.

**Figure 5 jcm-08-01352-f005:**
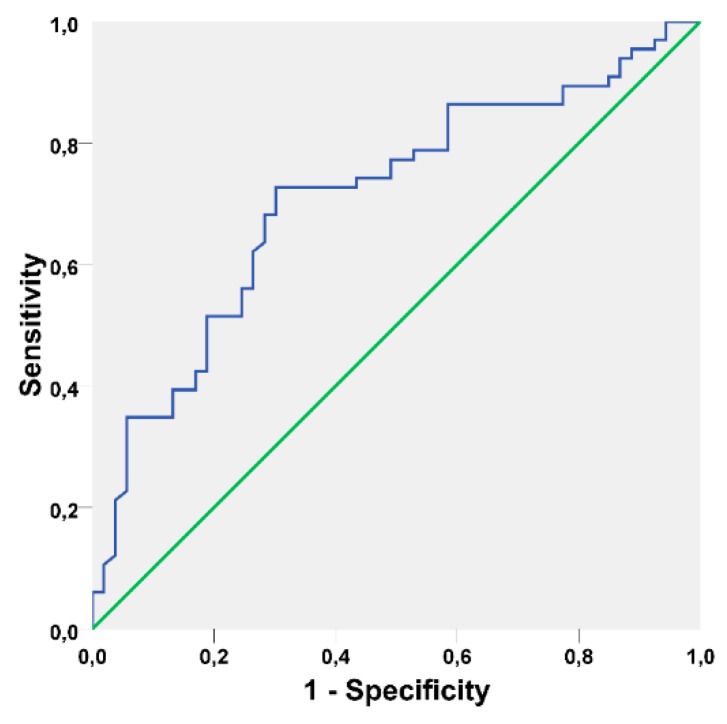
Receiver operating characteristic curve illustrating the performance of the prognostic model on the validation set (Area under the curve = 0.709 95% CI [0.616; 0.803], standard error = 0.048, *p* < 0.001).

**Table 1 jcm-08-01352-t001:** Sample characteristics in the derivation and validation sets of patients with Peri-implant disease (total, moderate disease and severe disease thresholds). Percentages are indicated according to columns.

Variables	Derivation Set (%)			Validation Set (%)		
	Total Sample	Moderate Disease	Severe Disease	Total Sample	Moderate Disease	Severe Disease
Demographics						
Number of patients	120 (100%)	54 (45%)	66 (55%)	120 (100%)	62 (51.7%)	58 (48.3%)
Average age (standard deviation)	58.6 (10.5)	56.7 (9.8)	60 (10.9)	58.6 (10.8)	58.6 (11.5)	58.6 (10.1)
Gender						
Female	73 (60.8%)	31 (57.4%)	42 (63.6%)	69 (57.5%)	28 (45.2%)	41 (70.7%)
Male	47 (39.2%)	23 (42.6%)	24 (36.4%)	51 (42.5%)	34 (54.8%)	17 (29.3%)
Implant status						
Survival	55 (45.8%)	31 (57.4%)	24 (36.4%)	53 (44.2%)	31 (50.8%	22 (37.9%)
Failure	65 (54.2%)	23 (42.6%)	42 (63.6%)	66 (55.0%)	30 (49.2%)	36 (62.1%)
History of Periodontitis						
Absence	29 (24.2%)	14 (25.9%)	15 (22.7%)	30 (25.0%)	16 (25.8%)	14 (24.1%)
Presence	91 (75.8%)	40 (74.1%)	51 (77.3%)	90 (75.0%)	46 (74.2%)	44 (75.9%)
Systemic comorbidities						
Absence	79 (65.8%)	35 (64.8%)	44 (66.7%)	80 (66.7%)	41 (66.1%)	39 (67.2%)
Presence	41 (34.2%)	19 (35.2%)	22 (33.3%)	40 (33.3%)	21 (33.9%)	19 (32.8%)
Smoking						
Non-smoker	77 (64.2%)	37 (68.5%)	40 (60.6%)	81 (67.5%)	42 (67.7%)	39 (67.5%)
Smoker	43 (35.8%)	17 (31.5%)	26 (39.4%)	39 (32.5%)	20 (32.3%)	19 (32.8%)
Proximity of implants/teeth						
Absence	86 (71.7%)	42 (77.8%)	44 (66.7%)	93 (77.5%)	48 (77.4%)	45 (77.6%)
Presence	34 (28.3%)	12 (22.2%)	22 (33.3%)	27 (22.5%)	14 (22.6%)	13 (22.4%)
Type of implant surface						
Machined	5 (4.2%)	2 (3.7%)	3 (4.5%)	7 (5.8%)	4 (6.5%)	3 (5.2%)
Anodically oxidized	115 (95.8%)	52 (96.3%)	63 (95.5%)	113 (94.2%)	58 (93.5%)	55 (94.8%)
Implant position per arch						
Maxilla	66 (55%)	29 (53.7%)	37 (56.1%)	65 (54.2%)	21 (33.9%)	34 (58.6%)
Mandible	54 (45%)	25 (46.3%)	29 (43.9%)	55 (45.8%)	41 (66.1%)	24 (41.4%)
Implant position in the arch						
Anterior	49 (40.8%)	21 (38.9%)	28 (42.4%)	42 (35.0%)	22 (35.5%)	20 (34.5%)
Posterior	71 (59.2%)	33 (61.1%)	38 (57.6%)	78 (65.0%)	40 (64.5%)	38 (65.5%)
Time of follow-up at diagnosis *						
≥4 years	69 (57.5%)	28 (51.9%)	41 (62.1%)	65 (54.2%)	37 (59.7%)	28 (48.3%)
<4 years	51 (42.5%)	26 (48.1%)	25 (37.9%)	55 (45.8%)	25 (40.3%)	30 (51.7%)
Implant length						
≤13 mm	62 (51.7%)	27 (50%)	35 (53%)	58 (48.3%)	28 (45.2%)	30 (51.7%)
>13 mm	58 (48.3%)	27 (50%)	31 (47%)	62 (51.7%)	34 (54.8%)	28 (48.3%)
Abutment height						
No abutment	11 (9.2%)	8 (14.8%)	3 (4.5%)	7 (5.8%)	2 (3.2%)	5 (8.6%)
1 mm	27 (22.5%)	13 (24.1%)	14 (21.2%)	26 (21.7%)	12 (19.4%)	14 (24.1%)
2 mm	36 (30.0%)	13 (24.1%)	23 (34.8%)	36 (30.0%)	22 (35.5%)	14 (24.1%)
3 mm	27 (22.5%)	10 (18.5%)	17 (25.8%)	23 (19.2%)	7 (11.3%)	16 (27.6%)
4 mm	15 (12.5%)	6 (11.1%)	9 (13.6%)	24 (20.0%)	17 (27.4%)	7 (12.1%)
5 mm	3 (2.5%)	3 (5.6%)	0 (0%)	4 (3.3%)	2 (3.2%)	2 (3.4%)
6 mm	1 (0.8%)	1 (1.9%)	0 (0%)	0 (0.0%)	0 (0%)	0 (0%)
Type of rehabilitation						
Single tooth	35 (29.2%)	18 (33.3%)	17 (25.8%)	37 (30.8%)	16 (25.8%)	21 (36.2%)
Partial	15 (12.5%)	7 (13%)	8 (12.1%)	10 (8.3%)	6 (9.7%)	4 (6.9%)
Full-arch	70 (58.3%)	29 (53.7%)	41 (62.1%)	73 (60.8%)	40 (64.5%)	33 (56.9%)
Type of material used in the restoration						
Full-ceramic	12 (10.0%)	9 (16.7%)	3 (4.5%)	8 (6.7%)	6 (9.7%)	2 (3.4%)
Metal-ceramic	52 (43.3%)	19 (35.2%)	33 (50.0%)	48 (40.0%)	25 (40.3%)	23 (39.7%)
Metal-acrylic resin	44 (36.7%)	17 (31.5%)	27 (40.9%)	44 (36.7%)	22 (35.5%)	22 (37.9%)
Full-acrylic resin	12 (10.0%)	9 (16.7%)	3 (4.5%)	20 (16.7%)	9 (14.5%)	11 (19.0%)
Type of opposing dentition						
Removable prosthesis	1 (0.8%)	0 (0%)	1 (1.5%)	2 (1.7%)	2 (3.2%)	0 (0%)
Natural teeth	57 (47.5%)	29 (53.7%)	28 (42.4%)	61 (50.8%)	34 (54.8%)	27 (46.6%)
Fixed prosthesis over natural teeth	1 (0.8%)	0 (0%)	1 (1.5%)	3 (2.5%)	2 (3.2%)	1 (1.7%)
Implant supported fixed prosthesis	61 (50.8%)	25 (46.3%)	36 (54.5%)	54 (45.0%)	24 (38.7%)	30 (51.7%)
Biofilm						
Absence	24 (20%)	16 (29.6%)	8 (12.1%)	24 (20%)	9 (14.5%)	15 (25.9%)
Presence	96 (80%)	38 (70.4%)	58 (87.9%)	96 (80%)	53 (85.5%)	43 (74.1%)
Bleeding						
Absence	17 (14.2%)	8 (14.8%)	9 (13.6%)	20 (16.7%)	8 (12.9%)	12 (20.7%)
Presence	103 (85.8%)	46 (85.2%)	57 (86.4%)	100 (83.3%)	54 (87.1%)	46 (79.3%)
Probing pocket depth in millimeters						
≤6 mm	108 (90%)	51 (94.4%)	57 (86.4%)	107 (89.2%)	53 (85.5%)	54 (93.1%)
>6 mm	12 (10%)	3 (5.6%)	9 (13.6%)	13 (10.8%)	9 (14.5%)	4 (6.9%)
Bone level at diagnosis						
Implants’ coronal third	54 (45%)	54 (100%)	0 (0%)	62 (51.7%)	62 (100%)	0 (0%)
Implants’ middle third	66 (55%)	0 (0%)	66 (100%)	58 (48.3%)	0 (0%)	58 (100%)
Mechanical complications						
Absence	115 (95.8%)	52 (96.3%)	63 (95.5%)	115 (95.8%)	59 (95.2%)	56 (96.6%)
Presence	5 (4.2%)	2 (3.7%)	3 (4.5%)	5 (4.2%)	3 (4.8%)	2 (3.4%)
Therapeutic approach						
Non-surgical	100 (83.3%)	48 (88.9%)	52 (78.8%)	97 (80.8%)	55 (88.7%)	42 (72.4%)
Surgical	20 (16.7%)	6 (11.1%)	14 (21.2%)	23 (19.2%)	7 (11.3%)	16 (27.6%)

* Follow-up of the derivation set: Average (standard deviation): 60 months (±32 months); range: 13 to 162 months. Follow-up of the validation set: Average (standard deviation): 62 months (±31 months); range: 14 to 162 months.

**Table 2 jcm-08-01352-t002:** Univariable and multivariable logistic regression analysis for bad prognosis of Peri-implant disease.

Variables	Relative Risk (RR) Crude Values (95% CI)	*p*-Value	Relative Risk (RR) Adjusted Values (95% CI) ^a^	*p*-Value	Beta Value
Age	1.04 (1.01; 1.08)	*p* = 0.030	1.04 (0.99; 1.08)	*p* = 0.105	0.035
Gender (male)	0.71 (0.34; 1.48)	*p* = 0.357			
History of Periodontitis	3.56 (1.46; 8.68)	*p* = 0.005	3.13 (1.15; 8.55)	*p* = 0.026	1.142
Systemic complications	1.13 (0.53; 2.40)	*p* = 0.760			
Smoking	1.49 (0.70; 3.18)	*p* = 0.302			
Proximity of implants/teeth	0.67 (0.30; 1.49)	*p* = 0.327			
Implant surface (machined)	5.02 (0.54; 46.31)	*p* = 0.155			
Implant position (maxilla)	1.36 (0.66; 2.80)	*p* = 0.408			
Implant position (posterior)	1.08 (0.52; 2.24)	*p* = 0.840			
Early disease development (<4 years)	2.45 (1.16; 5.20)	*p* = 0.019	3.99 (1.62; 9.82)	*p* = 0.003	1.383
Implant length (>13 mm)	2.84 (1.35; 5.99)	*p* = 0.006	3.52 (1.48; 8.37)	*p* = 0.004	1.258
Abutment height		*p* = 0.539			
No abutment	1.0 (reference)				
1 mm	2.48 (0.54; 11.40)				
2 mm	2.98 (0.68; 13.09)				
3 mm	4.53 (0.97; 21.14)				
4 mm	5.33 (0.97; 29.39)				
5 mm	-				
6 mm	-				
Type of rehabilitation		*p* = 0.112			
Full-arch	1.0 (reference)				
Single tooth	0.59 (0.26; 1.35)				
Partial	0.31 (0.10; 1.02)				
Type of material used in restoration		*p* = 0.287			
Full-ceramic	1.0 (reference)				
Metal-ceramic	0.57 (0.16; 2.02)				
Metal-acrylic resin	1.25 (0.34; 4.59)				
Acrylic resin	1.0 (0.20; 5.07)				
Type of opposing dentition		*p* = 0.660			
Implant-supported prosthesis	1.0 (reference)				
Removable prosthesis	-				
Natural teeth	0.63 (0.30; 1.30)				
Fixed prosthesis natural teeth	-				
Biofilm	2.33 (0.92; 5.86)	*p* = 0.071			
Bleeding	2.46 (0.84; 7.16)	*p* = 0.099			
Probing pocket depth (>6 mm)	2.79 (0.72; 10.85)	*p* = 0.140			
Disease severity (severe disease-bone level implants’ middle 1/3)	2.36 (1.13; 4.93)	*p* = 0.022	3.26 (1.37; 7.81)	*p* = 0.008	1.183
Mechanical complications	1.28 (0.21; 7.97)	*p* = 0.790			
Therapeutic approach (surgical)	1.71 (0.63; 4.66)	*p* = 0.290			

^a^*p* < 0.001 (Omnibus test); *p* = 0.350 (Hosmer and Lemeshow test); *R*^2^ = 0.315; logit (*p*) = −4.584 + 0.035 (age in life years) + 1.142 (history of periodontitis) + 1.383 (early disease development—less than 4 years of follow-up) + 1.258 (implant length > 13 mm) + 1.183 (severe peri-implant disease status—bone level on implants’ middle third).

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
