# Peer review of "A Prognostic Model for the Outcome of Nobel Biocare Dental Implants with Peri-Implant Disease after One Year"

_jcm, 2019, doi:10.3390/jcm8091352_

Round 1

Reviewer 1 Report

This is a well written paper using elegant statistical prognostic modelling to predict ailing dental implants' one year treatment outcome.  The effort to come up with reliable predictive formula that can be used in treatment decision making of ailing dental implant is very intersting and certainly of great interest.  However , the generalization of such formula would be difficult because of vast differences in implant macro and micro design.

It seems authors only included Nobel Biocare implants in this study ; however, the article attempts to generalize such findings across all dental implant brand and designs.  In fact, I suggest this fact to be incorporated in the title.

I was wondering if the variable of implant mobility also was measured , and if an objective device like Osstell or Periotest was used for such measurements?

Author Response

This is a well written paper using elegant statistical prognostic modelling to predict ailing dental implants' one year treatment outcome.  The effort to come up with reliable predictive formula that can be used in treatment decision making of ailing dental implant is very intersting and certainly of great interest.  However , the generalization of such formula would be difficult because of vast differences in implant macro and micro design.

Response: The authors thank the Reviewer’s effort in reviewing the manuscript. The authors agree with the Reviewer’s indication and placed the limitation on the Discussion section.

Changes: Discussion section, lines 345-347.

It seems authors only included Nobel Biocare implants in this study ; however, the article attempts to generalize such findings across all dental implant brand and designs.  In fact, I suggest this fact to be incorporated in the title.

Response: The authors thank the Reviewer’s suggestion. Besides the inclusion of the information as a limitation in the Discussion section the authors also introduced the information on the manuscript’s title as suggested by the Reviewer.

Changes: Title.

I was wondering if the variable of implant mobility also was measured , and if an objective device like Osstell or Periotest was used for such measurements?

Response: The authors thank the Reviewer’s query. The variable implant mobility was measured but manually. The authors introduced the information on the manuscript and included the fact that neither the Osstell or Periotest devices were used as a limitation on the Discussion section.

Changes: Materials and Methods section, line 118; Discussion section, lines 339-341

Reviewer 2 Report

This is an interesting study, however minor corrections are suggested prior to publication:

1. Concerning the experience of authors in Implant Dentistry, I am wondering why two important risks of periimplant disease were not included at the study: presence/absence of attached gingiva around implant and cemented vs. screwed restorations. This should be explained at the discussion section.

2. Please explain how 0.25N during periodontal probe was calibrated.

3. Please substitute the term "bacterial plaque" for "biofilm".

4. Results, Participants: if 1229 patients were eligible for inclusion and 27 were not included, why did the authors opt to analyze only 240 patients instead of 1202? And how was the randomization of implants selection performed?

5. Please start the first sentence of Discussion section with the most interesting findings of the study. Another introduction about what was done on the study is not necessary. Be more objective.

I am looking forward to read the results of re-evaluation at 5-years and 10-years post-diagnosis.

Author Response

This is an interesting study, however minor corrections are suggested prior to publication:

Concerning the experience of authors in Implant Dentistry, I am wondering why two important risks of periimplant disease were not included at the study: presence/absence of attached gingiva around implant and cemented vs. screwed restorations. This should be explained at the discussion section.

Response: The authors thank the Reviewer’s query. The authors agree with the Reviewer that both presence/absence of attached gingiva around the implant and cemented vs. screwed restorations may be important risk indicators for peri-implant disease (which in this case would mean to study implant factors prior to the incidence of disease). However, considering that the study was performed on implants with peri-implant disease and the evaluation was about the prognosis of those diseased implants, the variables would provide limited information as for example, the presence/absence of attached gingiva might be manipulated during a surgical intervention to attempt and treat the implant; while if cemented restorations or loosening of prosthetic components were the issue for the incidence of peri-implant disease, these would have been addressed on the treatment phase and therefore would have also limited value for prognosis of diseased implants. It was on this basis, together with the limited number of variables that could have been studied, that the variables were not included. The authors acknowledge the limitation of not including these two variables on the assessment on the Discussion section as suggested by the Reviewer.

Changes: Discussion section, lines 341-345.

Please explain how 0.25N during periodontal probe was calibrated.

Response: The authors thank the Reviewer’s query. The periodontal probes have a system (denominated “click” system) that automatically calibrates the pressure to 0.25N (Click-Probe, Kavo Kerr, Bioggio, Switzerland). For the avoidance of doubt, this information was introduced in the manuscript.

Changes: Materials and Methods section, line 146.

Please substitute the term "bacterial plaque" for "biofilm".

Response: The authors thank the Reviewer’s suggestion. The term “bacterial plaque” was replaced by the term “biofilm” as suggested by the Reviewer.

Changes: Introduction section, line 49; Materials and Methods section, line 143, 144, 160, Table 1 (line 220), Table 2 (line 259); Discussion section, line 321, 399, 400, 403, 402, 407, 430, 431, 432, 433, 436.

Results, Participants: if 1229 patients were eligible for inclusion and 27 were not included, why did the authors opt to analyze only 240 patients instead of 1202? And how was the randomization of implants selection performed?

Response: The authors thank the Reviewer’s query. The reduction of the data set was related to the sample size calculation. As this study is part of a PhD thesis, one of the outcomes evaluated was the capacity to a good use of resources (time and data) and also the capacity to produce results with the needed number of patients, in an era where big data negatively influences the analysis of the outcomes as the tendency for small differences to become significant associations dramatically increase in large data pools due to excess power, opposite to the problem that science used to be a victim of (reduced sample size and reduced power). This way, the challenge was to perform an investigation complying with these two parameters.

Changes: None.

Please start the first sentence of Discussion section with the most interesting findings of the study. Another introduction about what was done on the study is not necessary. Be more objective.

I am looking forward to read the results of re-evaluation at 5-years and 10-years post-diagnosis.

Response: The authors thank the Reviewer’s indication. The Discussion section was adapted as suggested. The authors will continue to evaluate the outcomes to assess the prediction ability to long-term post-diagnosis, thank you.

Changes: Discussion section, lines 301-305

Reviewer 3 Report

The aim of the manuscript entitled “A prognostic model for the outcome of dental implants with peri-implant disease after one year” was to establish and validate a prognostic model for implants diagnosed with periimplant disease.

The idea is interesting and the study is well done, however, a more useful approach was published by the same group of authors in February, this year: de Araújo Nobre, Miguel et al. “A Peri-Implant Disease Risk Score for Patients with Dental Implants: Validation and the Influence of the Interval between Maintenance Appointments.” Journal of clinical medicine vol. 8,2 252. 17 Feb. 2019.

Nevertheless, some drawbacks need to be addressed:

In the inclusion criteria: (rows 86-87) “1 year post-surgery (cut off value to prevent overlapping between early implant failure and peri-implant disease)”  could be better changed with 1 year post loading   (for the above-mentioned reason). The loading protocol was not mentioned in the considered risk factors. Is there any patient with immediate or early loading? What type of surgical protocol was used for implant insertion: submerged, not submerged? Are the restorations on implants fixed or removable, screw retained or cemented? The residual cement could be one of the risk factors for initial occurrence of peri-implantitis.

Author Response

The aim of the manuscript entitled “A prognostic model for the outcome of dental implants with peri-implant disease after one year” was to establish and validate a prognostic model for implants diagnosed with periimplant disease.

The idea is interesting and the study is well done, however, a more useful approach was published by the same group of authors in February, this year: de Araújo Nobre, Miguel et al. “A Peri-Implant Disease Risk Score for Patients with Dental Implants: Validation and the Influence of the Interval between Maintenance Appointments.” Journal of clinical medicine vol. 8,2 252. 17 Feb. 2019.

Response: The authors thank the Reviewer’s comment. The previous manuscript published in February 2019 concerns the study of factors that could represent risk indicators for peri-implant disease: This meant that the factors were evaluated before the incidence of peri-implant disease (healthy implants). The current manuscript concerns the study of potential prognosis (good or poor) of implants with peri-implant disease: This means that the factors were evaluated only on dental implants already with peri-implant disease.

Changes: None.

Nevertheless, some drawbacks need to be addressed:

In the inclusion criteria: (rows 86-87) “1 year post-surgery (cut off value to prevent overlapping between early implant failure and peri-implant disease)”  could be better changed with 1 year post loading   (for the above-mentioned reason). The loading protocol was not mentioned in the considered risk factors. Is there any patient with immediate or early loading? What type of surgical protocol was used for implant insertion: submerged, not submerged? Are the restorations on implants fixed or removable, screw retained or cemented? The residual cement could be one of the risk factors for initial occurrence of peri-implantitis.

Response: The authors thank the Reviewer’s comment. The authors agree with the Reviewer and introduced the term “one year post-loading” and also the information that all implants were restored with fixed prosthetic rehabilitations to avoid doubt. Concerning the effect of the residual cement, again the authors agree with the Reviewer that the residual cement could be one of the risk indicators for the initial occurrence of peri-implant disease, however, its prognostic value could be limited as the issue would have been dealt with on the peri-implant disease treatment phase (as the issue of loosening of prosthetic components would have been dealt with for screw retained restorations). Nevertheless, this was acknowledged in the Discussion section and included as a study limitation.

Changes: Materials and Methods section, lines 87,88; Discussion section, lines 341-345.

Round 2

Reviewer 2 Report

All suggestions and corrections were properly addressed.